# 4D imaging reveals mechanisms of clay-carbon protection and release

Judy Q. Yang [1], Xinning Zhang[2,3], Ian C. Bourg [3,4✉] & Howard A. Stone [1✉]

Soil absorbs about 20% of anthropogenic carbon emissions annually, and clay is one of the key carbon-capture materials. Although sorption to clay is widely assumed to strongly retard the microbial decomposition of soil organic matter, enhanced degradation of clay-associated organic carbon has been observed under certain conditions. The conditions in which clay influences microbial decomposition remain uncertain because the mechanisms of clay-organic carbon interactions are not fully understood. Here we reveal the spatiotemporal dynamics of carbon sorption and release within model clay aggregates and the role of enzymatic decomposition by directly imaging a transparent smectite clay on a microfluidic chip. We demonstrate that clay-carbon protection is due to the quasi-irreversible sorption of high molecular-weight sugars within clay aggregates and the exclusion of bacteria from these aggregates. We show that this physically-protected carbon can be enzymatically broken down into fragments that are released into solution. Further, we suggest improvements relevant to soil carbon models.

[1] Department of Mechanical and Aerospace Engineering, Princeton University, Princeton, NJ 08544, USA. [2] Department of Geosciences, Princeton University, Princeton, NJ 08544, USA. [3] Princeton Environmental Institute, Princeton University, Princeton, NJ 08544, USA. [4] Department of Civil and Environmental Engineering, Princeton University, Princeton, NJ 08544, USA. ✉email: bourg@princeton.edu; hastone@princeton.edu

Soil constitutes a vast carbon reservoir that exchanges around 60 gigatons of carbon annually with the atmosphere and absorbs about 20% of anthropogenic carbon emissions[1–3]. Consequently, variations in the capacity of soils to store carbon have tremendous impacts on the global carbon cycle and future climate[4,5]. This sensitivity of global climate to soil carbon storage presents both an obvious risk and a potentially powerful carbon mitigation tool[6].

The abundance of certain clays and clay minerals (in particular, smectites and nano-crystalline iron and aluminum oxides) is widely recognized as a key factor controlling the amount of carbon stored in soil and its release rate[7–9], yet the detailed processes responsible for this mineral protection remain unclear, especially in the presence of soil microbes and extracellular enzymes that degrade organic matter[1,5,10]. One of the common hypotheses is that sorption of organic molecules to clay surfaces temporarily protects this carbon from microbial decomposition[11–13]. Consequently, recently developed global and field-scale soil carbon models include a protected carbon pool determined by clay abundance and represented as a reversible sorption process[5,14]. This conceptual view, however, is challenged by evidence showing that clay-associated carbon remains remarkably labile and can be released in a short period of time (days), likely due to microbial and enzymatic activity, if low molecular-weight sugars (e.g. within root exudates) are input into the soil[15–17]. This phenomenon, hereinafter referred to as priming, suggests that microbial and extracellular enzymatic activity may directly impact the efficacy of mineral protection, in contrast with extant soil carbon models that represent mineral protection and biotic processes as independent and uncorrelated phenomena[5,18].

Direct observations of the interactions between clay, carbon, microorganisms, and extracellular enzymes (exoenzymes) are needed to improve global soil carbon and climate predictions and to enable effective designs of soil-based climate mitigation strategies[10,15–17]. Such observations are scarce due to a lack of real-time technology to visualize carbon dynamics within clay micro-aggregates, where most organic matter is stored[15,19,20]. Because of the opaque nature of clay micro-aggregates, observations of organic carbon within these aggregates have typically relied on destructive techniques that provide only a static snapshot of carbon–clay associations, i.e., a 2D or 3D image at a single point of time[21–23]. While these static snapshots provide useful information regarding the pore structure and the carbon distribution within clay aggregates, they provide only indirect insight into the dynamics of clay–carbon protection and release processes represented by soil carbon models[10].

Here, we demonstrate how carbon is stored in clay micro-aggregates and later released by exoenzymes in a model soil-on-a-chip[24]: a microfluidic device containing water, clay, and organic molecules designed to approximate organic sorption in soil. For the first time, we achieved four-dimensional (4D, three spatial dimensions plus time), imaging of carbon and bacteria within and surrounding clay aggregates by combining fluorescently labeled carbon and microbes with a transparent synthetic smectite clay (laponite) and confocal microscopy. With this 4D imaging method, we investigate the sorption of sugars with different molecular weights to smectite clay and show that high molecular-weight sugars (polysaccharides) are quasi-irreversibly sorbed within clay aggregates, i.e., we observe essentially no desorption on the time-scale of our experiments. In contrast, low molecular-weight sugars such as glucose are reversibly sorbed to clay. We find that this sorption creates a spatial separation, hence likely a physical protection, between the complex organic matter sorbed within clay aggregates and microorganisms confined to the periphery. Finally, we study how exoenzymes can rapidly promote the release of protected carbon. Based on our measurements, we propose an integrated view of how clay, bacteria, and exoenzymes together affect soil carbon storage and release and suggest an improved model structure for soil carbon predictions.

## Results

**Quasi-irreversible sorption.** The sorption and desorption of fluorescently labeled organic matter within micron-size clay aggregates, made from the synthetic and transparent smectite clay laponite, were imaged in 4D in a microfluidic channel using a confocal microscope, as shown in Fig. 1a–c. The micron-size clay aggregates were formed by suspending the 25 nm diameter by 1 nm thick laponite particles in buffer solutions (see the "Methods" section). Each clay aggregate (Fig. 1c), with length scale on the order of 10–100 μm, consists of thousands of nanometer-size primary laponite particles. The clay aggregates were first injected into the channel through a syringe by hand at a relatively high flow rate, i.e., several mL per minute (see the "Methods" section for details). After the clay aggregates deposited randomly in the channel (Fig. 1b), a solution containing dissolved fluorescently labeled organic matter was injected into the channel for a finite duration, long enough for the sorption to reach equilibrium (sorption phase), followed by injection of the pure buffer solution without organic matter (desorption phase). The sorption time is on the scale of hours, which resembles natural several-hour-long rainfall events. The clay aggregates were static during the sorption/desorption process because the flow was controlled by a syringe pump at a low flow rate (1 mL/h).

Two fluorescent organic compounds were investigated in this study: the fluorescein isothiocyanate (FITC) 3–5 kDa dextran and the fluorescence-labeled 2-deoxy-glucose (2-NBDG). We choose dextran, a polysaccharide derived from the monomer glucose, and the monosaccharide glucose as the representative soil organic matter because sugars are the most abundant components of organic matter in soil[25]. We focus on studying the impact of molecular weight on soil carbon sorption and desorption because molecular weight is one of the most important parameters used to characterize soil carbon[1]. Other details of the molecular structure of organic matter, such as the types of functional groups[1], that also affect sorption to clay are not considered in this study.

The fluorescence intensity of carbon within each clay aggregate was scanned using a confocal microscope with 1–2 μm horizontal and vertical resolutions (see the "Methods" section). An example of a 3D scan showing the diffusion of fluorescent 3–5 kDa dextran into one clay aggregate during the sorption process is shown in Fig. 1c. The 4D images, or the time sequence of the 3D images, are shown in Supplementary Movies 1 and 2. The average fluorescence intensity within the clay aggregates scales linearly with the concentration of carbon within the aggregates as shown by the calibration experiments with fluorescently labeled dextran of known concentration (Supplementary Fig. 2). The average fluorescence intensity within each clay aggregate was first calculated (see the "Methods" section). Afterwards, the mean and standard error of the average fluorescence intensity of multiple clay aggregates in one microfluidic channel (e.g. Fig. 1b) were calculated as a function of time as shown by the symbols and error bars in Fig. 1d, e. During flow of solution containing the 3–5 kDa dextran, the average fluorescence intensity within the clay aggregates increased and reached equilibrium within about 5 h (Fig. 1d and Supplementary Movie 1). Upon subsequent flow of organic-free solution, the average fluorescence intensity decreased slowly and the majority of the dextran (>1/2 of the average fluorescence intensity) remained within the clay aggregate after about 50 h, indicating that this higher molecular-weight sugar was largely irreversibly sorbed under the conditions of our measurements. Note that spikes in the fluorescence intensity

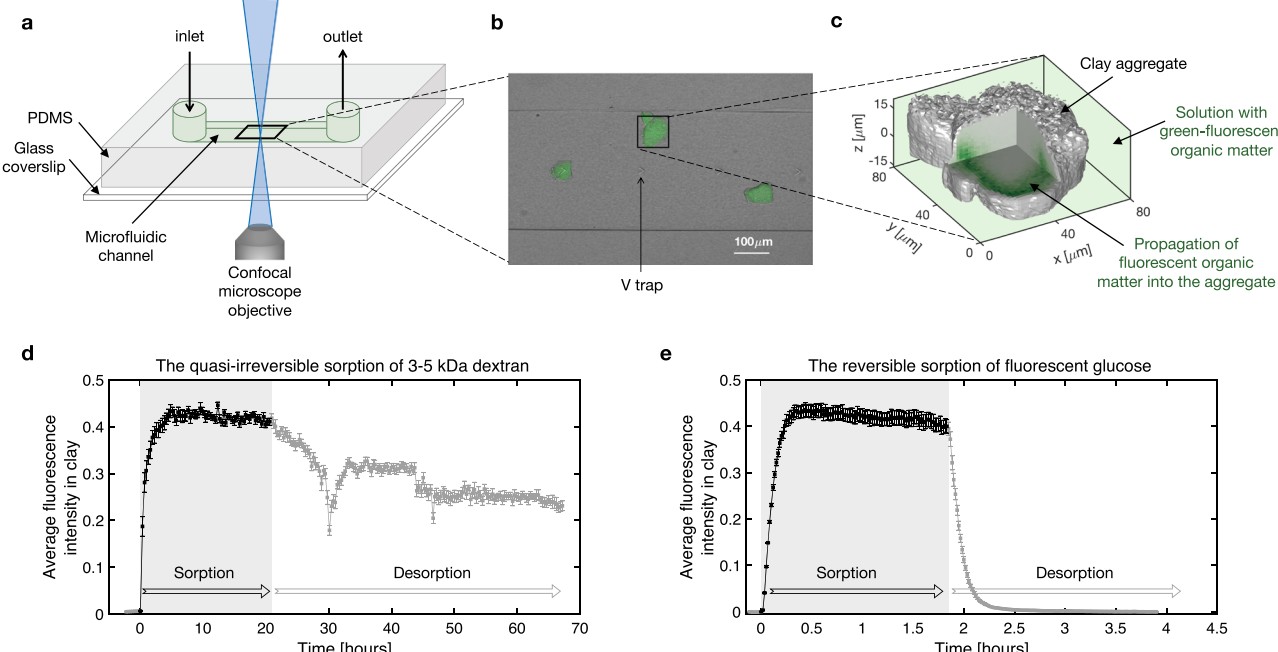

**Fig. 1 4D imaging of higher and lower molecular weight fluorescent organic matter in clay reveals two types of clay–carbon sorption. a** Schematic diagram of the experimental set up. The microfluidic channel was made from polydimethylsiloxane (PDMS) and a glass cover slip (see the "Methods" section for details). **b** Cross-sectional image of the 300 μm wide by 40 μm high microfluidic channel. A fluorescent image was superimposed on a bright-field image. Three clay aggregates sorbed with green fluorescent carbon (3–5 kDa dextran) appear green. Note that the tiny V-symbols in the middle of the channel were originally designed to trap the clay aggregates. However, these clay traps were found unnecessary because most clay aggregates with sizes larger than the channel depth remained stuck and static in the channel during the flow (1 mL/h), regardless of the traps (see the "Methods" section for details). **c** 3D snapshot of the clay aggregate indicated by the black box in (**b**). The clay aggregate is indicated by the silver color and fluorescent carbon is indicated by the green color (see the "Methods" section for details). At $t = 0$ h, a flow with carbon (high molecular-weight dextran or low molecular-weight glucose) was injected into the channel and the sorption of dextran or glucose to clay is shown as black symbols with error bars in (**d**) and (**e**), respectively. The symbol and the error bar represent the mean and standard error of the average fluorescence intensity within multiple aggregates in one microfluidic channel. Specifically, three aggregates as shown in (**b**) were used for the 3–5 kDa dextran (**d**), and five aggregates (Supplementary Fig. 7a) were used for the fluorescent glucose (**e**). After the sorption of carbon to clay reached equilibrium, a flow without carbon was injected into the channel, and the desorption of carbon from clay is shown as gray symbols with error bars in (**d**) and (**e**). The sorption and desorption curves suggest that the sorption of high molecular-weight sugars to clay is quasi-irreversible on a time-scale of days (**d**), while the sorption of low molecular-weight sugars to clay is readily reversible (**e**).

curve, such as the one at round 30 h, represent ambient noise due to external factors. In contrast, the sorption of the low molecular-weight (340 Da) fluorescent glucose was fully and rapidly reversible (Fig. 1e and Supplementary Movie 2). Replicate experiments with different random arrangements of the clay aggregates and sorption times (Supplementary Figs. 5–7) consistently show similar types of sorption, i.e., quasi-irreversible sorption of dextran and reversible sorption of glucose, suggesting that the arrangement of clay aggregates, which may affect the rate of sorption, does not affect the type of sorption.

Our experimental results differ from an existing popular conceptual model of soil organic matter persistence[1], which suggests that smaller molecular size carbon has greater opportunity for mineral protection (e.g. Fig. 2 in ref. [1]). Furthermore, the quasi-irreversible sorption of the 3–5 kDa dextran and the reversible sorption of 340 Da fluorescent glucose together suggest that, at least in the case of sugars, quasi-irreversible sorption of organic matter to clay only occurs for high molecular-weight compounds, as observed here for compounds with molecular weight ≥3 kDa.

**Mechanisms for quasi-irreversible sorption.** The quasi-irreversible sorption of high molecular-weight sugars to smectite clay observed in Fig. 1 was further examined by measuring the equilibrium sorption isotherm of 3–5 kDa dextran on clay in well-mixed batch-reactor conditions (i.e., the equilibrium

relationship between organic matter concentration in solution, $C_e$, and sorbed to clay, $C_s$). The sorption isotherm was obtained by mixing clay and dextran solution in glass vials and measuring the dextran concentration in the solution before and after sorption (see the "Methods" section). Our results, shown in Fig. 2a, indicate that the concentration of adsorbed dextran was essentially invariant with its concentration in solution for $C_e$ values ranging over two orders of magnitude (0.009–2 g/L) before increasing for $C_e > 2$ g/L following a classical S-shaped sorption isotherm[26]. The plateau value of $C_s$ for $C_e < 2$ g/L, $C_{s\text{-plateau}} = 0.2 \pm 0.1$ g/g, is consistent with previous observations of a plateau loading of organic matter on smectite clay in sorption experiments and in marine sediments[11,27]. The plateau concentration $C_{s\text{-plateau}} = 0.2 \pm 0.1$ g/g is quantitatively consistent with a monolayer coating on the entire clay surface. Specifically, if the value of $C_{s\text{-plateau}}$ is expressed as an average thickness of sorbed dextran on the clay surface, $h = C_{s\text{-plateau}}/a_s\rho_{\text{dextran}}$, where the specific surface area of smectite[28] $a_s = 760 \pm 40$ m²/g and the density of dextran[29] $\rho_{\text{dextran}} \sim 1.3 \pm 0.3$ g/cm³, then $h = 2.1 \pm 1.2$ Å is approximately the known thickness of a monolayer of water or graphite (3 Å). As shown in Supplementary Fig. 4, adsorption experiments with higher molecular-weight dextran showed the same plateau loading, whereas experiments with low molecular-weight sugar (glucose) were consistent with a linear adsorption isotherm with no plateau.

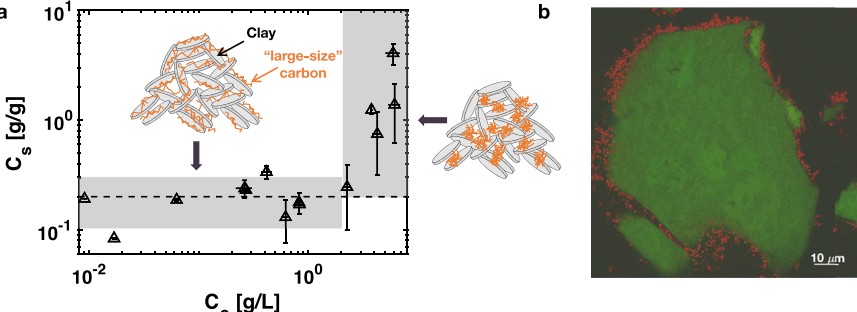

**Fig. 2 Clay–carbon sorption isotherm and bacterial–clay interactions. a** The sorption isotherm of 3–5 kDa dextran on clay suggests two sorption mechanisms. Each data point represents one batch sorption experiment, i.e., one experiment with 10 mg of clay added to 10 mL solution in a glass vial (see the "Methods" section). For each batch experiment, three drops were sampled from the solutions after the sorption and three sets of $C_e$, organic matter concentration in solution, were estimated from the fluorescence intensity measurements. Afterwards, three sets of $C_s$, organic matter concentration sorbed to clay, were calculated from $C_e$ and the dextran concentration in solution before sorption (see the "Methods" section). The triangles and error bars represent the mean and the standard error of these three sets of measurements. **b** A typical confocal microscopy image of the distribution of red fluorescent soil bacteria, *Pseudomonas aeruginosa*, around clay micro-aggregates loaded with green-fluorescent glucose.

The sharp increase in $C_s$ for $C_e > 2$ g/L (Fig. 2a) indicates secondary sorption of high molecular-weight organic matter in addition to the monolayer-equivalent sorption noted above. We hypothesize that this secondary sorption reflects the formation of an organic-rich phase on the clay surface, either through capillary condensation in mesopores within the clay-aggregates, e.g. analogous to the condensation of water vapor in mesoporous media including clays at high humidity[30], or through a surface-promoted aggregation of organic matter at the clay–water interface, e.g. analogous to the promotion of surfactant hemi-micelles at solid–water interfaces. This hypothesis is consistent with observations of sub-micron discrete organic patches in natural soils[31,32].

**Bacteria and exoenzymes.** Next, we investigated the impacts of soil bacteria and extracellular enzymes on carbon protection within clay micro-aggregates. We first showed that bacteria were confined to the periphery of clay aggregates and thus were spatially separated from organic matter sorbed within clay aggregates. Specifically, we incubated a bacterium found widely in soils, *Pseudomonas aeruginosa*, with the transparent clay in a nutrient solution containing regular glucose as a carbon source in a culture well dish (see the "Methods" section). After one day of incubation, the clay–bacteria mixture was imaged under a confocal microscope (Fig. 2b and Supplementary Fig. 8). The bacteria appeared red because a red fluorescent gene, mCherry, was engineered on the bacterial chromosome[33]. The clay appeared green due to the addition of green fluorescent glucose as a tag. The cross-sectional image shows that almost no bacteria penetrated inside the clay aggregates (Fig. 2b), confirming the hypothesis that micron-size bacteria cannot penetrate into the nanometer-size pores of clay[10,34]. The exclusion of bacteria from the clay, consistently observed in over four replicate experiments (e.g. Fig. 2b and Supplementary Fig. 8), in effect, physically protects the organic matter from direct contact with soil bacteria, in agreement with the observed correlations between soil organic carbon and smectite clay abundance in temperate soils[7–9].

Further, we demonstrated that the release of clay-bound carbon, observed in field and laboratory experiments[15–17], can be explained by exoenzymes produced by some soil bacteria and fungi. Here we added a commercially available dextranase (an enzyme that breaks down dextran by cleaving glycosidic linkages between its saccharide components) to dextran sorption experiments. We chose to use the extracted dextranase because the soil

bacterium used in the present experiments, *P. aeruginosa*, does not produce dextranase. Solutions containing dextrans with different molecular weights (3–5 kDa, 20 kDa, 70 kDa), followed by a solution containing the enzyme dextranase, were injected sequentially into a microfluidic device designed to mimic a soil macropore, cavities with typical pore size larger than 50–100 μm (Fig. 3). To represent the intermittency of natural flows, e.g. rainfall events with several-hour duration, the injections were modified at well-spaced hour-scale time intervals. The clay emitted green and red light when green and red fluorescently labeled dextrans were sorbed to it, respectively. The average green and red fluorescence intensities, represented by the green and red curves, were calculated in the immobile clay regions (the three black contours in Fig. 3a). During the injection of green 3–5 kDa dextran and red 20 kDa dextran, the average green and red intensities of clay increased, respectively, indicating a kinetically limited uptake of 3–5 and 20 kDa dextrans in clay aggregates. Specifically, after the injection of the red 20 kDa dextran stopped, only the edge of the clay appeared red (Fig. 3b, c). This is because the diffusivity of the red 20 kDa dextran in the clay was small such that only the edge of the clay was exposed to the red dextran, i.e., the sorption of the red dextran did not last long enough for the sorption in the clay regions to saturate or reach equilibrium (see Supplementary Discussion in the Supplementary Information file for discussion about the diffusivity). Only the area sorbed with red dextran at the end of this sorption stage was used to calculate the average red fluorescence intensity in clay. During the injection of the green 70 kDa dextran, the changes in the average green and red intensities within the clay were not noticeable, suggesting that the diffusion of 70 kDa dextran into clay was too slow to be observed within the experimental time frame. Afterwards, when the buffer solution without any organic matter was injected into the channel, no changes in average green and red intensities of clay were observed, confirming that the sorption of high molecular-weight sugars (≥3 kDa) to clay is quasi-irreversible on the time-scale of our experiments.

Upon addition of dextranase, however, the average green and red intensities within the aggregates rapidly (within several hours) decreased to almost zero. This result indicates that, unlike micron-size bacteria, nanometer-size exoenzymes such as the 7 nm dextranase (Supplementary Fig. 1), can penetrate into the clay aggregates and break down sorbed dextrans into smaller fragments for which sorption is reversible, which leads to a rapid release of carbon into solution. Replicate experiments with only one type of high molecular-weight carbon and the exoenzyme

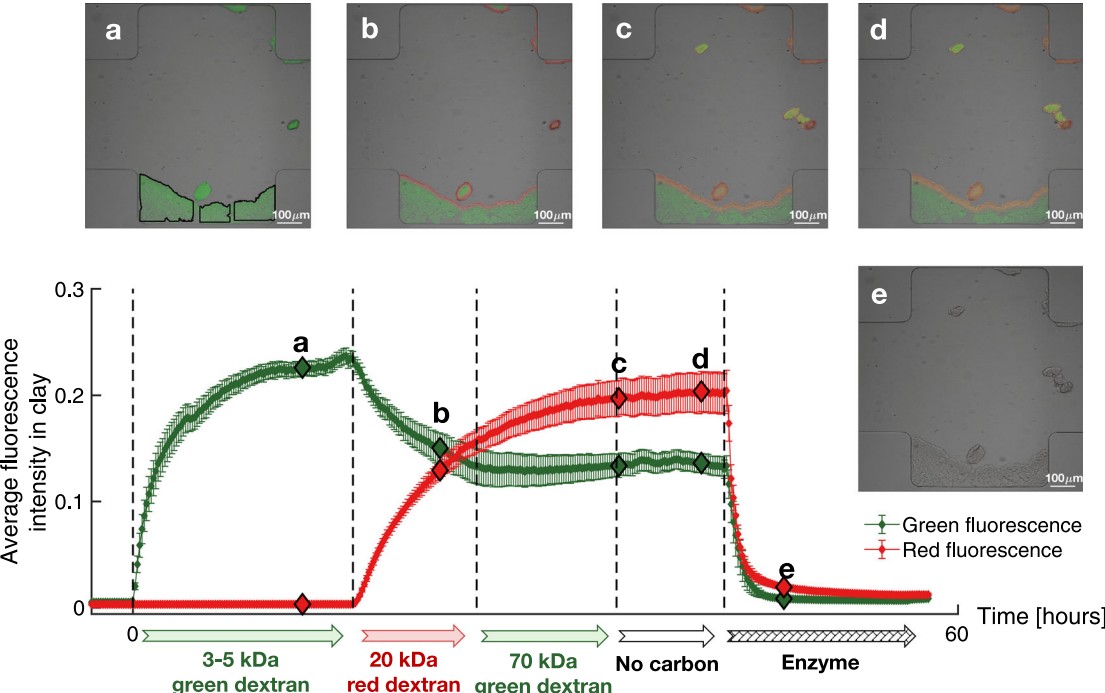

**Fig. 3 Exoenzyme dextranase releases irreversibly sorbed high molecular-weight carbon.** Cross-sectional fluorescence images of a microfluidic channel designed to mimic a soil macropore (typical pore size >50–100 μm) containing clay micro-aggregates, taken at times **a**–**e** (diamond symbols along the intensity curves), super-imposed on bright-field images. The three black contours in panel **a** show the three immobile clay regions used to calculate the mean and the standard errors of the average fluorescence intensity in clay. Background buffer solutions variably enriched with organic substances were injected sequentially into the channel: (1) 0.05 g/L green 3–5 kDa dextran, (2) 0.05 g/L red 20 kDa dextran, (3) 0.05 g/L green 70 kDa dextran, (4) no organics, and (5) 2 g/L enzyme (dextranase). The symbols and error bars represent the means and the standard errors of the average fluorescence intensity within the three immobile clay regions (outlined by three black contours in Fig. 3a). Note that during the experiment, three clay aggregates appeared in the field of view from the upstream of the channel (compare panels **b** and **c**). This is because when we switched the valve to change the injection to a different flow/solution, small vibrations of the inflow tubes mobilized the clay aggregates. Such temporal disturbances do not affect the result because only the intensity in the immobile clay regions was considered. The start time and duration of each injection are indicated, respectively, by the vertical dashed lines and the arrows underneath the horizontal time axis.

dextranase consistently show the desorption of carbon immediately after the injection of exoenzymes (e.g. Supplementary Fig. 10). Our results are consistent with the observation that the enzyme glucose oxidase retains most of its catalytic activity after being sorbed within smectite clay aggregates[35]. The breakdown of high molecular-weight organic matter by exoenzymes within clay aggregates is further demonstrated by the fluorescence intensity profiles of dextrans in clay (Supplementary Figs. 9 and 10), which show a rapid resorption of fragments of dextrans broken down by the exoenzyme dextranase (see Supplementary Discussion in the Supplementary Information file).

## Discussion
Building on the above observations, we propose an integrated conceptual model for interactions between clay, carbon, microbes, and exoenzymes (Fig. 4a). First, organic matter of various sizes (300 Da–2 MDa, Supplementary Table 1) can diffuse into clay aggregates and sorb to clay. The sorption of low molecular-weight carbon (<3 kDa) is reversible, thus it can diffuse out of clay where it is consumed by microbes. In contrast, high molecular-weight carbon (≥3 kDa) is quasi-irreversibly sorbed to clay, with a plateau loading equivalent to a monolayer on the entire clay surface. As bacteria are confined to the periphery of clay aggregates, sorbed carbon is physically protected from direct contact with bacteria. Nevertheless, exoenzymes such as dextranase, produced by some bacteria, can diffuse into clay aggregates, where they breakdown high molecular-weight carbon, causing release into

solution where the carbon can potentially be utilized by surrounding bacteria.

The resulting conceptual model reconciles observations of mineral protection[7–9] and priming[15–17], i.e., the intensified loss of clay-protected carbon following addition of low molecular-weight sugars. On the one hand, we demonstrate that clay protects organic matter through physical separation from soil bacteria. On the other hand, we reveal that high molecular-weight sugars are particularly strongly sorbed, but can be broken down by exoenzymes within clay aggregates and released into solution. These findings are consistent with the observed decrease of clay-protected carbon in priming experiments[15–17]. Specifically, when low molecular-weight carbon is added to soil, some exoenzyme-producing bacteria become more active and produce more exoenzymes. As exoenzymes diffuse into clay aggregates, they break down high molecular-weight organic compounds into smaller fragments that are readily released into solution, becoming available to surrounding bacteria, some of which produce yet more exoenzymes. This positive feedback loop, which we expect to be modulated by the diversity of bacteria, exoenzymes, and carbon forms in nature, can lead to enhanced degradation of clay-protected soil carbon and corresponding rapid emission of greenhouse gases, as observed in priming[15–17].

Our results also suggest that it may be possible to directly observe priming using the soil-on-a-chip methodology developed here, by replacing the exogenous dextranase used in Fig. 3 with enzyme-producing bacteria. Note that in this study the bacteria and clay interactions were observed in a culture dish; the next

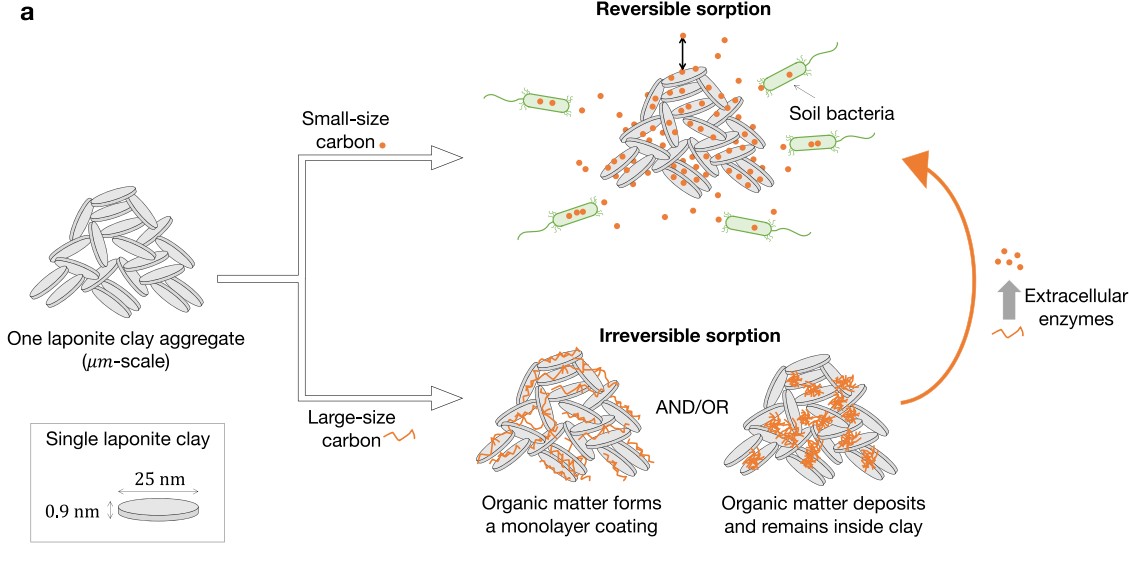

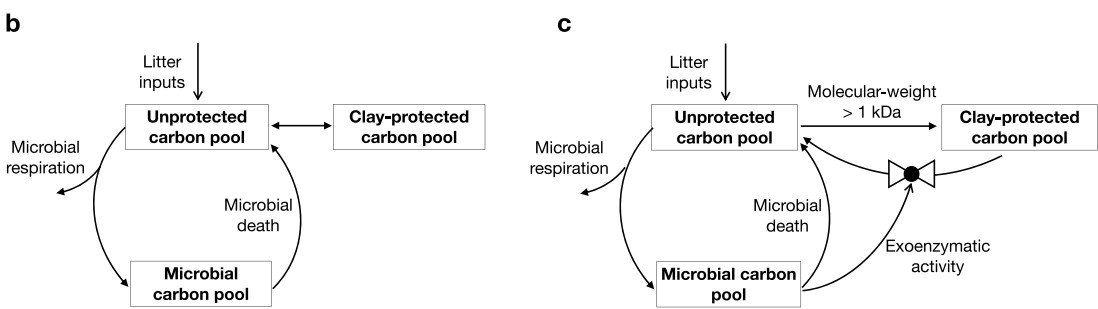

**Fig. 4 Proposed conceptual model for soil carbon interaction with clay minerals and implications for soil carbon models. a** An integrated conceptual model for interactions between clay, carbon, microbes, and exoenzymes is proposed. Note: drawing not to scale. **b** A representative soil carbon model structure, which implements clay–carbon protection and biotic activity as distinct processes[5,18]. **c** A revised soil carbon model that implements clay–carbon protection and biotic activity as coupled processes. The release of the clay-protected carbon could be modeled as a function of exoenzymatic activity, which is likely a function of the bacterial biomass and growth rate.

step to mimic a soil more closely would be to directly incubate bacteria in a microfluidic channel with clay. Further, note that water content and oxygen level also impact soil carbon dynamics[5,36]. While here we use a water-saturated microfluidic setup with gas-permeable PDMS walls, future microfluidic experiments could examine the impact of water saturation and oxygen limitations on microbial respiration in microfluidic devices.

A key outcome of the present study is the demonstration that microbial and extracellular enzymatic activity can directly impact the efficacy of mineral protection. However, many representative soil carbon models implement biotic activity and mineral protection as distinct processes[5,18], as shown in Fig. 4b. Based on the findings in this paper, we suggest an improved soil carbon model structure that treats biotic activity as a direct cause of the release of clay-associated organic carbon, as indicated by the pathway with a step controlled by exoenzymatic activity in Fig. 4c. In addition, the release of high molecular-weight carbon from clay by exoenzymes suggests that future research investigating the activity and diversity of exoenzymes in soils and the interactions of these enzymes with minerals may be particularly important for predicting the fate of soil carbon.

## Methods

**Clay, background buffer solution, and enzyme**. The laponite clay powder used here is Laponite-RD purchased from BYK USA Inc. A single primary laponite particle is a 25-nm-diameter by 1-nm-thick disk-like crystal (BYK manual).

Micron-size clay aggregates were created by mixing the clay powder with a background buffer solution using a vortex mixer at 200 rpm for about 1 min. The buffer solution is M9 minimum medium without carbon (48 mM $Na_2HPO_4$, 22 mM $KH_2PO_4$, 9 mM NaCl, 19 mM $NH_4Cl$, 2 mM $MgSO_4$, 0.1 mM $CaCl_2$), supplemented with micronutrients (0.03 μM $(NH_4)_6(Mo_7)_{24}$, 4 μM $H_3BO_3$, 0.3 μM $CoCl_2$, 0.1 μM $CuSO_4$, 0.8 μM $MnCl_2$, 0.1 μM $ZnSO_4$, and 0.1 μM $FeSO_4$). The average pore size of the laponite clay is around 2 nm[37]. The measured pH of the solution is 7. The organic matter used in this study is shown in Supplementary Fig. 1 and listed in Supplementary Table 1. The fluorescent glucose, 2-NBDG (2-(N-(7-Nitrobenz-2-oxa-1,3-diazol-4-yl)Amino)-2-Deoxyglucose), was purchased from Thermo Fisher Scientific (Catalog number: N13195). The FITC dextrans were purchased from Sigma (CAS Number 60842-46-8). The enzyme used in this study is dextranase produced by *Penicillium* sp. (a fungi) and purchased from Sigma (CAS Number 9025-70-1).

**Microfluidic clay sorption and desorption experiments**. The microfluidic channels used in this study, such as the ones shown in Figs. 1 and 3, were fabricated using soft lithography[38]. To fabricate a microfluidic channel, we first made a mold for the channel on a SU-8-coated silicon wafer using the laser pattern generator Heidelberg microPG101 (Princeton PRISM Center). Afterwards, we made the channel from the mold by pouring polydimethylsiloxane (PDMS) on the molded silicon wafer. The PDMS was made by mixing 10-parts base elastomer and 1-part curing agent (Dow Sylgard 184 Silicone Elastomer Kit purchased from Ellsworth Adhesives). After the PDMS was cured in a 60 °C oven for around 12 h, we removed the PDMS from the molded silicon wafer and punched holes at the channel inlet and outlet using a 1 mm hole puncher. Then, we bonded the PDMS to a #1.5 cover glass after treating the two bonding surfaces using a PE-25 Venus plasma cleaner.

The height of all the channels used in this study is 40 μm and the width of the straight channel is 300–400 μm. The length of the channel from inlet to outlet is about 5 mm. The detailed geometries of the channels shown in Figs. 1 and 3 are

shown in Supplementary Fig. 11a, b, respectively. Note that the V-shaped traps in Fig. 1b and Supplementary Fig. 11a were originally designed to trap clay, however, these traps were found to be unnecessary because the clay aggregates were immobilized in the channels due to the narrow depth of the channel instead of the traps. Therefore, these traps were not used in the microfluidic chamber shown in Fig. 3. The main function of the microfluidic channel is to hold the clay micro-aggregates in place during exposure to fluid flow, thus our results should be reproducible in any straight microfluidic channel, as long as the channel depth and width are similar to our design (40 μm depth and 300–400 μm width).

For the sorption and desorption experiment, 10 mg laponite clay powder was mixed in 10 mL background solution using a vortex mixer at 2000 rpm for about 1 min. Then, the clay solution was stored at room temperature for 24 h to make sure that the clay aggregates swelled to equilibrium. Afterwards, the clay solution was injected into the microfluidic channel through a syringe by hand at a flow rate on the order of mL/min. Note that during the manual injection, aggregates smaller than channel depth (40 μm) were carried away by the flow, while the larger ones were squeezed into the channel and many of them became immobilized in the channel. Next, background solutions with 0.05 g/L organic matter and without organic matter were injected sequentially into the channel using a syringe pump operating at a flow rate of 1 mL/h. During this constant flow, most of the clay aggregates that were trapped in the channel during the manual injection remained static due to the relatively low flow rate (1 mL/h) of the experiment compared with the manual injection (several mL/min).

**Confocal microscopy**. The fluorescence intensity inside clay aggregates was scanned using confocal microscopy (Leica TCS SP5) at 1.5 μm-horizontal and 1 μm-vertical resolution for the sorption/desorption experiments (Figs. 1 and 3). 2 μm vertical resolution was used for the enzyme release experiment (Fig. 3) to reduce file size. One typical image represents one horizontal scan with around 512 by 512 pixels, and a 3D image is a stack of around 40 images, representing scans at 40 vertical positions. The cross-sectional images shown in Fig. 1b and other figures were cropped for visual clarity. The average fluorescence intensity in each aggregate was scanned at 1-min intervals for the sorption/desorption experiments in Fig. 1 and at 10-min intervals for the exoenzyme release experiment in Fig. 3. Only a subset of the data was shown in Figs. 1 and 3 for visual clarity of the figure. The background intensity at $t = 0$ h was subtracted from both curves. The slow decrease in average dextran fluorescence intensity in clay after $t = 50$ h is likely due to photobleaching of the fluorescent molecules. Additional evidence of photobleaching and its dependence on experimental duration and scanning frequency is provided in Supplementary Figs. 5 and 6.

For experiments with only green fluorescent organic matter (2-NBDG and FITC dextrans), the laser used for excitation has wavelength 488 nm and the emission filter covers 498–633 nm. To visualize *P. aeruginosa* PA14 mCherry[33], a 543 nm laser was used and the emission filter ranged from 553 to 689 nm. The bacterial culture image was scanned at 0.12 μm by 0.12 μm resolution (Fig. 2b). To visualize the red organic matter (TRITC dextran), the laser wavelength was 543 nm and the emission filter was 571–582 nm. When green fluorescent organic matter was visualized with red fluorescent organic matter and bacteria, the emission filter of the green signal was narrowed to 498–520 nm to avoid overlap with the red signal.

**Image analysis**. During the sorption and desorption processes, the fluorescence intensity in clay first increased to a plateau and then decreased (e.g. Fig. 1d, e). The outer surface of each clay aggregate was constructed from the fluorescence intensity when the clay was saturated with fluorescent carbon, i.e., when the average fluorescence intensity reached maximum. The fluorescence intensity was rather uniformly distributed within clay aggregates at saturation (with average intensity between 0.2 and 0.5), such that we define the clay outer surface as the location where the fluorescence intensity drops to around 0.1. Visual comparison with bright-field images show that our method accurately constructed the outer surface of the clay aggregates. The average fluorescence intensity of each clay aggregate was then estimated as the average fluorescence intensity of the pixels enclosed by the outer surface of the clay aggregate. Figure 1d shows the mean and standard error of the average fluorescence intensity within the three clay aggregates shown in Fig. 1b. Figure 1e presents the mean and the standard error of the average fluorescence intensity in five aggregates enclosed by the red boxes in Supplementary Fig. 7a. Because the fluorescence intensity was quite uniformly distributed in each clay aggregate, running averages calculated with different numbers of aggregates show that three clay aggregates are enough to capture the mean intensity in the clay aggregates. The same method was used to identify the outer surface of clay aggregates in other experiments (e.g. Fig. 3).

**Sorption isotherm experiments**. To measure the sorption isotherm of organic compounds on laponite clay (Fig. 2), samples of 10 mg clay powder were added to 10 mL background solutions containing 3–5 kDa dextran at different concentrations in 20 mL glass vials. The glass vials were then placed in a box wrapped with aluminum foil to avoid photobleaching. For each dextran concentration, a sample without clay was prepared in identical conditions as a control. The box was then placed in a 200 rpm shaker for 3 days at room temperature. Afterwards, the supernatant solution was centrifuged at $1350 \times g$ for 10 min. The average

fluorescence intensity of the solution before and after adding clay was measured and compared with calibrated average fluorescence intensity versus dextran concentration curves (Supplementary Fig. 2), from which we calculated $C_o$ (mass/volume), the dextran concentration in the solution before adding the clay, and $C_e$, the dextran concentration in the supernatant solution after the sorption of dextran to clay reached equilibrium. The concentration of sorbed dextran in clay (mass/mass) was calculated by mass balance as $C_s = (C_o − C_e)V/M_{clay}$, with $V$ and $M_{clay}$ representing the volume of the solution and the mass of clay, respectively. To test whether the average fluorescence intensity of the clay correlated with the amount of organic matter sorbed to clay $C_s$, clay slurries recovered by centrifugation were transferred to a #1.5 glass slip and imaged under a confocal microscope (Supplementary Fig. 2).

**Bacterial strain and bacteria–clay culture experiments**. The soil bacterial strain used here, *P. aeruginosa* PA14 with mCherry fluorescent gene engineered into the chromosome, was provided by A. Siryaporn[33]. Bacteria were first grown overnight in standard lysogeny broth medium at 37 °C with 200 rpm shaking. After 10 mg clay was added to 1 mL background solution with 4 g/L D-glucose and 0.04 g/L 2-NBDG, 100 μL bacteria overnight culture was added to the clay mixture in a 35 mm culture well dish. The bacteria–clay mixture was then placed in a 37 °C incubator for 24 h before being imaged using a confocal microscope.

## Data availability
All data generated or analyzed during this study were extracted from the Supplementary Movies and are available from the corresponding authors on reasonable requests.

## Code availability
The codes for imaging processing are shared on GitHub: v1.0 JudyYang-umn/-SoilC_image_processing_codes v1.0 (DOI: 10.5281/zenodo.4304503).

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

## Acknowledgements

This research was supported by the Princeton Environmental Institute through the Grand Challenges program and the Carbon Mitigation Initiative. The authors thank J. Wilmoth and J. Yan for help with the bacteria-related experiments and J. Hong, H. Ma, S. Myneni, S. Pacala, A. Porporato, E. Shevliakova, and R. Socolow for insightful discussions.

## Author contributions

J.Q.Y., I.C.B., and H.A.S. conceived the study. J.Q.Y. designed and conducted the experiments with the help of I.C.B., H.A.S., and X.Z. J.Q.Y., I.C.B., and H.A.S. wrote the initial draft and all authors discussed the results and contributed to the writing.

## Competing interests

The authors declare no competing interests.
