## [Peer Review File · Nature Communications]

REVIEWER COMMENTS

Reviewer #3 (Remarks to the Author):

My comments have been satisfactorily addressed.

Yiqi Luo

Reviewer #5 (Remarks to the Author):

The present manuscript entitled "4D imaging reveals mechanisms of clay-protection and release" explores the processes driving the protection of carbon within mineral aggregates. For this purpose, authors use a specific clay mineral, laponite, together with low molecular weight (glucose) and high molecular weight (dextran) sugars. The main outcome of this study is that the high molecular weight dextran stays irreversibly bind to the clay, while low molecular weight glucose is easily desorbed from the clay particles. The further application of exoenzymes releases both kinds of sugar in the solution. This original study provides interesting outcomes on the mechanisms driving carbon storage in soils. Although numerous additional factors need to be considered to tackle the soil complexity (e.g., indigenous organic matter and microorganisms, interaction between microbial communities, minerals of various nature), this study presents a promising methodological advance that deserves to be considered.

The authors have addressed correctly the reviewers' comments.

I have a few additional minor comments and thoughts:

Abstract

L14: Clay is not „the key“, but rather „one of the key“ carbon-capture material. Indeed, there are numerous other processes that enable carbon storage in soils. To me, for example, microorganisms are an important "carbon-capture material" considering the microbial loop concept.

L18: Clay itself does not inhibit microbial decomposition, but rather the interaction between clay and OM.

Introduction

L36: Do authors consider aluminium oxides as clay minerals?

L39-40: Another emerging hypothesis is that a significant amount of carbon, especially in hotspots such as the rhizosphere is directly taken up by microorganisms in the vicinity of the carbon source. Figure 1: Please explain what is PDMS in the caption Fig. 1a. Please highlight the "V" symbol on the channel in fig. 1b. It is barely visible on the figure.

L101: I would be curious to see what would happen if the aggregate already contained indigenous organic matter as it is the case in natural soils.

Results

L103: At this point of the manuscript, it is difficult to understand what is a "transparent clay aggregates". I would simply say a "clay aggregate" or explain more.

L121-122: What do authors mean by "other details"? Please be more precise and add a reference to this assumption.

L137-138: How do authors explain the phenomenon happening at 30 hours for dextran, when the intensity drops before increasing again?

L144: What if the glucose was first taken up by microorganisms, as it would most likely happen in a

natural soil?

L198: If I understood well from the material and methods, the bacteria were grown separately and the clay glucose mixture was prepared prior to the addition of bacteria, right? What were the reasons for not adding the glucose source after mixing the bacteria to the soil. Indeed, in a natural soil, the microorganisms would most likely be already present in the vicinity of the carbon source and might thus be able to intercept the carbon source before it diffuses within the aggregates. This is partly stated by the authors in the conclusion L317-319.

L215: I would not say "our soil bacterium" but rather "the soil bacterium used in the present experiment".

Conclusion

L317-322: I would suggest authors to place these limits earlier in the conclusion. This would enable ending on the potentials of the method rather than its limits.

The authors are grateful to Dr. Yiqi Luo (Reviewer #3) and the other two anonymous reviewers for their helpful comments. We have provided a detailed point-by-point reply to all comments below. The original comments are given in black font, and our responses are given in blue font. For Reviewer #2, the comments from the second round of review are given in red font and our responses are given in purple; comments from the first round of review that have been addressed were not copied here to avoid redundancy.

Reviewer #3 (Remarks to the Author):

My comments have been satisfactorily addressed.

Yiqi Luo

Thank you for your support. We are glad that we addressed all of your previous comments.

Reviewer #5 (Remarks to the Author):

The present manuscript entitled “4D imaging reveals mechanisms of clay-protection and release” explores the processes driving the protection of carbon within mineral aggregates. For this purpose, authors use a specific clay mineral, laponite, together with low molecular weight (glucose) and high molecular weight (dextran) sugars. The main outcome of this study is that the high molecular weight dextran stays irreversibly bind to the clay, while low molecular weight glucose is easily desorbed from the clay particles. The further application of exoenzymes releases both kinds of sugar in the solution. This original study provides interesting outcomes on the mechanisms driving carbon storage in soils. Although numerous additional factors need to be considered to tackle the soil complexity (e.g., indigenous organic matter and microorganisms, interaction between microbial communities, minerals of various nature), this study presents a promising methodological advance that deserves to be considered.

Thank you for your support. We are pleased to read that “this original study provides interesting outcomes on the mechanisms driving carbon storage in soils,” and that “this study presents a promising methodological advance that deserves to be considered.”

The authors have addressed correctly the reviewers’ comments.

Thank you. We are glad that you agree that we addressed the previous reviewers’ comments correctly.

I have a few additional minor comments and thoughts:

Abstract

L14: Clay is not „the key“, but rather „one of the key“ carbon-capture material. Indeed, there are numerous other processes that enable carbon storage in soils. To me, for example, microorganisms are an important “carbon-capture material” considering the microbial loop concept.

Thank you for this remark. We have revised the wording to “clay is one of the key carbon-capture materials” (lines 11-12).

L18: Clay itself does not inhibit microbial decomposition, but rather the interaction between clay and OM.

Thank you. We have now changed the word “inhibits” to “influences” (line 14).

Introduction

L36: Do authors consider aluminium oxides as clay minerals?

The term “clay” refers to the finest grain-size fraction in soils and sediments (typically < 2 micrometers). The phrase “clay minerals” is most often used to refer to phyllosilicate minerals, but is sometimes used more broadly to also include other clay-sized minerals (including aluminium and iron oxides). The laponite used in our study matches both definitions of “clay minerals”. To avoid giving the impression that we endorse one definition, we replaced “clay minerals” by “clays and clay minerals” (line 31).

L39-40: Another emerging hypothesis is that a significant amount of carbon, especially in hotspots such as the rhizosphere is directly taken up by microorganisms in the vicinity of the carbon source. Thank you for mentioning other hypotheses. To make it clearer, we have now changed our wording to “One of the common hypotheses” (line 35).

Figure 1: Please explain what is PDMS in the caption Fig. 1a. Please highlight the “V” symbol on the channel in fig. 1b. It is barely visible on the figure.

Thank you for pointing out these details. We have added the following sentences in the caption of Figure 1 to explain PDMS, which is also explained in the Methods section (lines 288-291).

“The microfluidic channel was made from polydimethylsiloxane (PDMS) and a glass cover slip (see Methods for details)” (lines 511-512).

For the “V” symbol, we have added an arrow that points to the “V” symbol in Fig. 1b. We have also added sentences to explain the “V” symbol in the caption (lines 514-518). We also added the exact details of the “V”-symbol in Supplementary Fig. 11(a).

L101: I would be curious to see what would happen if the aggregate already contained indigenous organic matter as it is the case in natural soils.

Thank you for the interesting question. We agree with the reviewer that the use of clay aggregates containing indigenous organic matter would be very interesting. Exogenous organic matter was used in our experiments to enable the use of fluorescently-labeled organic matter. We believe that our exogenous organic matter provides a reasonable approximation of indigenous organic matter in natural soils for two reasons. First, the irreversible sorption of large molecular-weight organic matter within clay aggregates likely resembles the incorporation of organic matter from natural sources, such as plant residues, into the clay aggregates in natural soils, thus we think the irreversibly-sorbed organic matter mimics indigenous organic matter. Second, the mechanism whereby extracellular enzymes can diffuse into and break down organic matter within clay aggregates should operate regardless of whether or not the organic matter is indigenous.

Results

L103: At this point of the manuscript, it is difficult to understand what is a “transparent clay aggregates”. I would simply say a “clay aggregate” or explain more.

Thank you for the suggestion. We have changed the wording to “clay aggregates, made from the synthetic and transparent smectite clay laponite” (lines 78-79).

L121-122: What do authors mean by “other details”? Please be more precise and add a reference to this assumption.

Thank you for the suggestion. We have changed the sentence to “Note that other details of the molecular structure of organic matter, such as the types of functional groups¹, also affect its sorption to clay and are not considered in this study” (lines 97-99).

L137-138: How do authors explain the phenomenon happening at 30 hours for dextran, when the intensity drops before increasing again?

The spike at 30 hours is ambient noise, likely due to factors such as voltage fluctuations. We have added the following sentence: “Note that spikes in the fluorescent intensity curve, such as the one at round 30 hours, represent ambient noise due to factors like voltage fluctuations” (line 117-119).

L144: What if the glucose was first taken up by microorganisms, as it would most likely happen in a natural soil?

Thank you for the interesting question. If the glucose was first taken up by microorganisms, then it should be used by the microorganisms and would not exist in the solution. Here we are discussing the situation where there is abundant glucose in the solution, in which case the small molecular-weight glucose would be reversibly-sorbed within the clay aggregates. The reversible sorption means that the clay-associated carbon acts like a carbon/food reservoir for microbes with a supply rate dictated by desorption and diffusion kinetics. We have related discussions in the first paragraph of the Discussion section (lines 225-235).

L198: If I understood well from the material and methods, the bacteria were grown separately and the clay glucose mixture was prepared prior to the addition of bacteria, right? What were the reasons for not adding the glucose source after mixing the bacteria to the soil. Indeed, in a natural soil, the microorganisms would most likely be already present in the vicinity of the carbon source and might thus be able to intercept the carbon source before it diffuses within the aggregates. This is partly stated by the authors in the conclusion L317-319.

Thank you for the interesting question. Yes, we chose to mix the clay and the solution with fluorescent glucose first before adding bacteria, because the focus of the experiment was on whether and how bacteria could access clay bound organic matter. Adding the bacteria with the glucose at the same time would lead to a more complex experiment in which bacteria would interact with free glucose prior to its sorption to clay and with free glucose desorbed from clay, with the results being more difficult to interpret.

L215: I would not say “our soil bacterium” but rather “the soil bacterium used in the present experiment”. Thank you for helping us improve our wording. We have changed the wording (line 185).

Conclusion

L317-322: I would suggest authors to place these limits earlier in the conclusion. This would enable ending on the potentials of the method rather than its limits.

Thank you for the suggestion. We have moved these limits to the end of the previous paragraph (lines 252-258). We think this rearrangement of text improved the ending of our paper and appreciate the suggestion.

Reviewer #2 (Remarks to the Author):

Review Report: 4D imaging reveals mechanisms of clay-carbon protection and release Yang et al. describe a microfluidic approach that allows researchers to gain visual assess to transparent synthesis smectite clay (liponite) aggregates and investigate clay-carbon interactions on the microscale. Direct observations of interactions between clay, carbon, microorganisms and enzymes are needed to improve current carbon models that consider mineral protection and biotic processes as independent and uncorrelated phenomena (despite evidence suggesting otherwise).

Although a simple method, this is – to the best of my knowledge – the first account that uses a microfluidic setup to investigate carbon dynamics with clay microaggregates in real-time. This

platform has a large potential to answer a variety of new and exciting biological questions. However, the work is unfortunately more promising than delivering. I have several concerns regarding the presented data as described below.

Thank you for your support. We are pleased to read that our study is “the first account that uses a microfluidic setup to investigate carbon dynamics with clay microaggregates in real-time. This platform has a large potential to answer a variety of new and exciting biological questions.” We have addressed your questions point by point below.

Major comments

- There is no information regarding the complete architecture of the microfluidic device design, therefore it would be impossible to reproduce this experiment. An overview of the device is required as a basic minimum (indicating channel/feature sizes), including specific details of protocols for photo- and soft lithography methods used to create the device implemented in this study etc. There is no information regarding how samples are injected (syringes used, interface, methods of injection etc.) and the justification for using this particular device design. It seems that these experiments could be done in a simple flow cell, for example.

Thank you for the excellent suggestion. We have now included a description of the experimental set up, one representative snapshot of the microfluidic cross-section, as well as one example clay aggregate in the revised Figure 1. The size of the channel is mentioned in the caption of Figure 1 and described in the Methods section. The procedures to make the device and inject clay aggregates into the channel are described in detail in the Methods section (lines 427-440) and now are also mentioned in the first paragraph of the Results section (e.g. lines 107-109, lines 113-115).

Device Architecture Point

- The authors have only provided a few very basic details about the microfluidic device used to collect the data in this manuscript.

- There is still no overview of the complete microfluidic channel architecture, and therefore it remains impossible to reproduce the device and hence the experiments detailed in this manuscript.

Thank you for the above two questions regarding the channel geometry. We have described the sizes of the channel geometry that we think are most relevant in the Methods section (lines 296-302). To make the microfluidic channel structures clearer, we have now added two figures showing the complete geometry of the microfluidic channels with labeled dimensions as shown in Supplementary Fig. 11 (shown below). The function of the channel is mainly to hold the clay aggregates in place during exposure to fluid flow, thus our results should be reproducible in any straight microfluidic channels, as long as the channel depth and width are similar to our design (40 μm depth and 300-400 μm width). We also added related descriptions in the manuscript (lines 299-305).

Supplementary Figure 11. The geometries of the microfluidic channels. (a) The geometry of the microfluidic channel used in Fig. 1 of the main text. The inset shows the geometry of the “V”-shaped clay trap in the middle of the channel. The space between the clay traps in the flow direction is $200\ \mu\text{m}$. (b) The geometry of the microfluidic channel used in Fig. 3 of the main text. The depth of both channels is $40\ \mu\text{m}$. The approximate field of the views of the confocal microscope are indicated by the blue dashed boxes.

- The few pictures in the manuscript that show microchannel architecture are unclear. For example, Figure 1b shows a straight channel with V-shaped pillars, then Figure 3 shows a channel that is apparently designed to mimic a macropore with certain parts of the channel have been widened. Is this one device? Or two devices even? This is confusing and misleading and the authors need to be open about the exact design used to meet basic ethical research reporting standards.

Yes, the designs of the microfluidic channels in Fig. 1 and Fig. 3 are different, i.e., they are two different devices. We have now added figures showing the complete structures of the two channels in Supplementary Fig. 11. We have referred the readers to the Supplementary Fig. 11 for the complete geometry of the channels in the Methods section (lines 299-302). Details about the “V”-shaped pillars/traps are discussed in our response to the next two comments.

- Line 86: “The tiny “V” symbols in the middle of the channel were designed to trap the clay” These were not mentioned previously. It is very difficult to see the V-shaped pillars. A zoom in/close up is required and details of how these are incorporated into the over microchannel architecture.

A detailed figure of the “V”-shaped traps is now shown in the inset of the Supplementary Fig. 11(a).

- The authors state that the V-shaped pillars were designed to trap the clay aggregates, but this does not seem to be very efficient (only 1 particle of 3 is trapped in the image displayed in Figure 1b). Can the authors comment on trapping efficiency of this device?

Thank you for the observation and question. The “V”-shaped traps were originally designed to trap the clay aggregates, but we found that these traps were not necessary for trapping the clay aggregates, which is why we removed these traps in the channel used in Fig. 3. Specifically, we found that most clay

aggregates with sizes larger than the channel depth remained stuck or static in the channel during the experiments with the flow rate 1 mL/hr, regardless of the traps. In fact, during the manual injection of clay aggregates into the channel, aggregates smaller than channel depth ($40\ \mu\text{m}$) were carried away by the flow, while the larger ones were squeezed into the channel and many of them got stuck in the channel. These aggregates were the ones we used in the experiments. We have added related discussions about the “V” traps and the reason clay were trapped in the channel in the caption of Fig. 1(b) and the Methods section (lines 296-305 and 311-318).

Methods Point

- With regards to reproducibility, the details provided in the methods section are extremely disappointing and the authors have not addressed this point. For example, I assume that by stating “the PDMS was made by mixing 10-parts base elastomer and 1-part curing agent”, the authors are referring to a Sylgard kit? There are many, which can influence the properties of the elastomeric polymer. Which one was used?

Thank you for pointing out the missing details about PDMS. Yes, the PDMS we used was the commonly used Dow Sylgard 184 Silicone Elastomer Kit purchased from Ellsworth Adhesives. We have now added the information in the Methods section (lines 289-291).

- How is the silicon wafer produced? This is a complicated process and no mention of how this was manufactured was included, i.e. what exact protocols were used to generate channel heights of $40\ \mu\text{m}$?

Thank you for pointing out the missing details about the microfluidic fabrication process. We realized that we did not document some details that we think are standard in microfluidic fabrication. We have now added these details as well as a reference to a review paper about soft lithography (lines 285-294). The microfluidic devices can easily be reproduced in any standard microfluidic fabrication lab or center, which should provide a protocol on how to fabricate SU-8 coating with different heights on a silicon wafer.

- How was the PDMS bonded to the cover glass?

Thank you for pointing out the missing PDMS bonding method. We have now added the following sentence in the Methods section: “We bonded the PDMS to a #1.5 cover glass after treating the two bonding surfaces using PE-25 Venus plasma cleaner” (lines 293-294).

- Details of equipment used (model, manufacturer) are not provided.

Thank you for the question. We have added the model and manufacturer of the soft lithography equipment in the Methods section (e.g., lines 287-288, 294).

- Lines 442-464: what type of confocal microscope was used?

Thank you for the question. We used a Leica TCS SP5 confocal microscope and have added the information (line 321-322).

- Etc etc etc....

Thank you for pointing out the missing technical details. We think we have now provided all the necessary information about the design and fabrication of our microfluidic devices. We added: “The main function of the microfluidic channel is to hold the clay micro-aggregates in place during exposure to fluid flow, thus our results should be reproducible in any straight microfluidic channel, as long as the channel depth and width are similar to our design ($40\ \mu\text{m}$ depth and $300\text{-}400\ \mu\text{m}$ width).” (lines 302-305)

- Are the clay aggregates fixed in microfluidic device, or do they stick randomly to the PDMS/glass surface? It seems that this is random and the clay aggregates can be introduced/removed by the flow. How do you control this and is this reproducible? How does this influence their experience of the

environment around them (i.e. is it uniform)? This needs clarification and characterization.

Thank you for the question. The clay aggregates were static in the microfluidic device during the sorption and desorption processes because the flow was controlled by a syringe pump at low flow rate (1mL/hour). The aggregates were injected into the microfluidic channel through a syringe by hand at relatively high flow rate (several mL/min). Details about how clay aggregates were static during the flow are described in the Methods section (lines 427-440) and now also mentioned in the first paragraph of the Results section (e.g. lines 107-109, lines 113-115).

- I am still not convinced that clay aggregates remain static in the microfluidic device. In Supplementary Movie 3, there is clearly movement of aggregates in the device after around 1-2 seconds, and also in Supplementary Movie 4 (around 4 seconds).

- Considering that it has now been declared there are V-shaped pillars in the channel design to trap clay aggregates, I do not see these in the movies.

Thank you for the above two questions related to the trapping of clay in the microfluidic channel and the “V” shaped trap. As we responded to some of the comments above, the clay aggregates remained immobilized in the channel during the experiments with flow rate 1 mL/hr because the clay aggregates had sizes larger than the channel depth. These clay aggregates were squeezed into the channel during the manual injection at a higher flow rate (several mL/min). The “V”-shaped traps were originally designed to trap the clay aggregates, but we found that these traps were not necessary for trapping clay so we removed them in later experiments. We have added related discussions about the method to trap clay and the “V”-shaped traps in lines 299-305 and 311-318. In addition, yes, there was some clay that got mobilized during the experiments; these clay aggregates were not considered in our calculation, as we have already mentioned in lines 560. We note that the aggregation, rheology, and transport properties of clay colloids remain incompletely understood despite a half-century of scientific examination as reviewed for example by Bailey et al. (2015) Smectite clay – inorganic nanoparticle mixed suspensions: phase behaviour and rheology, *Soft Matter*, 11:222. The optimal methodology for generating and trapping clay aggregates in a microfluidic device remains unknown and its precise determination would require an extensive research effort far beyond the scope of this manuscript.

- Towards the end of Supplementary Video 3 (13 s onwards), it seems that there is a bacterial contamination around the clay particles (there is an increase in mass). This is very concerning, as it indicates that the system is not controlled.

Our experiments are well controlled as described in the Methods section. The “bacteria” you mentioned at the end of Supplementary Video 3 are not bacteria, which should have much smaller size (on the order of 1 μm) with different shapes. Instead, they are clay aggregates from the upstream of the channel. The clay aggregates were static most of the time in the channel during the flow because the flow rate was low (1 mL/hour, see Methods). However, for this specific experiment, when we switched the valve to change the injection to a different flow/solution, some small vibrations of the tubes mobilized the clay aggregates in the upstream. Such temporary small disturbances do not affect our results because we only tracked the intensity within the immobile clay aggregates, which are indicated by the black contours in Figure 3(a). We have added related discussions in lines 250-254.

- I am still not convinced there is no bacterial contamination. If you look at the later frames of Supplementary Movie 3 there are 1000s of tiny little dots that appear all over the field of view (from around 14s onwards). These are definitely bacteria (unless you can prove otherwise). This coincides with the increase in mass around the clay aggregates to the right of the field of view. Additionally, the increase in mass is uniform around the aggregate. If the increase in mass is due to additional clay aggregates carried by flow, as you suggest, they would accumulate on one side of the clay only.

Thank you for the careful observation and the proposed explanation about the black dots around the clay aggregates to the right of the field of view, which occurred near the end of Supplementary Movie 3. The black dots occurred after the injection of the enzyme which may have produced some glucose, so we cannot completely exclude the possibility that these dots may be bacteria. However, our results and

interpretation will not be affected by any possible bacterial growth. First, we did not observe similar growing little dots in other videos with higher resolution, e.g., Supplementary Movie 1 and 2. Second, we do not have any bacteria that can degrade the large molecular-size dextrans in our lab (dextran degrading bacteria are not common), therefore there is no carbon sources for bacteria to grow in experiments with dextrans. For experiments with glucose, the sorption and desorption experiment only lasted for four hours and no growing black dots were observed in the video (e.g., Supplementary Movie 2). Furthermore, none of the four key messages of the manuscript (i.e., the types of sorption depend on molecular weight, the mechanisms of irreversible sorption, the exclusion of bacteria outside clay aggregates, and the release of clay-bound carbon by enzymes) would be affected by any potential bacterial growth. To be more cautious, we warned our readers of potential bacterial growth in the caption of Supplementary Movie 3 (lines 19-22 in the Supplementary information).

REVIEWERS' COMMENTS

Reviewer #2 (Remarks to the Author):

My comments have been addressed and I recommend for publication.

Reviewer #5 (Remarks to the Author):

Authors have addressed my comments correctly.

Reviewer #2 (Remarks to the Author):

My comments have been addressed and I recommend for publication.

Reviewer #5 (Remarks to the Author):

Authors have addressed my comments correctly.

The authors are grateful to the reviewers for their helpful comments, which have significantly improved our paper. We are glad to learn that the reviewers think our manuscript is now suitable for publication in Nature Communications.